# *TARTESSUS*: A Customized Electrospun Drug Delivery System Loaded with Irinotecan for Local and Sustained Chemotherapy Release in Pancreatic Cancer

**DOI:** 10.3390/bioengineering10020183

**Published:** 2023-02-01

**Authors:** Carmen Cepeda-Franco, Oihane Mitxelena-Iribarren, Francisco José Calero-Castro, Malen Astigarraga, Juan M. Castillo-Tuñon, Iman Laga, Sheila Pereira, Sergio Arana, Maite Mujika, Javier Padillo-Ruiz

**Affiliations:** 1Department of General Surgery, Virgen del Rocio University Hospital, 41013 Seville, Spain; 2Oncology Surgery, Cell Therapy, and Organ Transplantation Group, Institute of Biomedicine of Seville (IBiS), Virgen del Rocio University Hospital, University of Seville, 41013 Seville, Spain; 3CEIT-Basque Research and Technology Alliance (BRTA), 20008 Donostia-San Sebastián, Spain; 4Tecnun, Universidad de Navarra, 20018 Donostia-San Sebastián, Spain; 5Department of General Surgery, Virgen Macarena University Hospital, 41009 Seville, Spain

**Keywords:** electrospinning, nanofiber membrane, pancreatic cancer, drug delivery system, irinotecan

## Abstract

Post-surgical chemotherapy in pancreatic cancer has notorious side effects due to the high dose required. Multiple devices have been designed to tackle this aspect and achieve a delayed drug release. This study aimed to explore the controlled and sustained local delivery of a reduced drug dose from an irinotecan-loaded electrospun nanofiber membrane (named *TARTESSUS*) that can be placed on the patients’ tissue after tumor resection surgery. The drug delivery system formulation was made of polycaprolactone (PCL). The mechanical properties and the release kinetics of the drug were adjusted by the electrospinning parameters and by the polymer ratio between 10 w.t.% and 14 w.t.% of PCL in formic acid:acetic acid:chloroform (47.5:47.5:5). The irinotecan release analysis was performed and three different release periods were obtained, depending on the concentration of the polymer in the dissolution. The *TARTESSUS* device was tested in 2D and 3D cell cultures and it demonstrated a decrease in cell viability in 2D culture between 72 h and day 7 from the start of treatment. In 3D culture, a decrease in viability was seen between 72 h, day 7 (*p* < 0.001), day 10 (*p* < 0.001), 14 (*p* < 0.001), and day 17 (*p* = 0.003) as well as a decrease in proliferation between 72 h and day 10 (*p* = 0.030) and a reduction in spheroid size during days 10 (*p* = 0.001), 14 (*p* < 0.001), and 17 (*p* < 0.001). In conclusion, *TARTESSUS* showed a successful encapsulation of a chemotherapeutic drug and a sustained and delayed release with an adjustable releasing period to optimize the therapeutic effect in pancreatic cancer treatment.

## 1. Introduction

Pancreatic cancer is one of the most aggressive tumors with the worst prognosis [1] for those affected. This can be concluded from the fact that only 4% of patients have a 5-year survival rate after their diagnosis. The main reason for this is the difficulty in diagnosing this pathology since by the time observable symptomatology begins to develop, the tumor is already in an advanced state, which makes its treatment extremely difficult. The problem caused by the late diagnosis requires multidisciplinary treatments because only 20% of the tumors can be eliminated exclusively by conventional surgery [2]. Although surgery is considered to be the preferred strategy, the complete elimination of residual tumor cells and circulating tumor cells (CTCs) is challenging [3]. The 10-year survival rate of patients after surgery is still less than 4%, which is mainly related to the high recurrence rate caused by incomplete surgical resection and a lack of effective postoperative adjuvant therapy [4].

Conventional therapies such as surgery, chemotherapy, or radiotherapy cause side effects and are often insufficient on their own. Although it is now common to use a combination of two or more of them in parallel, new technologies or technological improvements to these treatments are required to increase the low survival rate and reduce the side effects [5]. Currently, surgery is usually combined with chemotherapy for this type of cancer. The most widely used chemotherapeutics are gemcitabine [6] and FOLFIRINOX, the latter consisting of three anti-cancer drugs (oxaliplatin, irinotecan, 5-fluorouracil) and a vitamin, leucovorin [7,8]. FOLFIRINOX, the most promising treatment for pancreatic cancer, extends the average survival to 11 months but has great side effects.

As an alternative to systemic adjuvant therapy, local drug delivery systems (DDSs) [5,9,10,11] have been designed to be implanted directly into the tumor bed after surgery and provide sustained drug release to the tumor site, which can overcome drug transport barriers and reduce side effects as well as improve patient adherence [12] DDSs are responsible for drug delivery in a specific way, either by attacking tumor cells directly and exclusively or by leading conventional drugs to the affected organ [11]. Some examples of DDSs are patches [13,14], hydrogels [15,16], nanoparticles, even using new carriers such as MCM-41, MCM-48, or SBA-15 [17,18,19,20,21], or electrospun meshes [8,9,22,23,24]. Compared to traditional systemic or regional chemotherapy, local administration of the drug using a drug-eluting scaffold reduces the dose required to achieve a comparable anti-cancer effect by approximately two-thirds [8].

For the development of degradable devices for internal areas requiring longer release times, as in this case, recent studies have demonstrated that electrospinning is a promising technique given the versatility of materials that can be used in the process, the ease of manufacture, the low cost, and the possibility of producing large quantities in a single process [25]. Moreover, the technology is still being updated to favor the different applications in which it is used [26]. These advantages, related to electrospinning as a manufacturing technique, make this method a versatile solution that provides an interesting approach for tackling not only non-clinical problems such as oil/water separation with nanofibrous membranes [27] but also more medical-related aspects like the loco-regional treatment of chronic diseases or tumors such as pancreatic cancer [26,28]. Regarding this last scenario, to avoid a re-intervention for the removal of the device from the inside of the patient, not only do biodegradable polymers need to be used, but the non-toxicity of the polymers also needs to be proved.

In recent years, several studies including the ones carried out by our group have analyzed the release of 5-fluorouracil (5-FU) and it has been shown that a controlled release of the drug can be obtained through a membrane formed by fibers [26,29,30,31]. Consequently, this study was based on procedures carried out with 5-FU, but using irinotecan this time, another component of FOLFIRINOX that acts as a prodrug and targets DNA topoisomerase I, an essential enzyme for DNA replication [32].

On the other hand, the DDSs studied thus far [5,9,10,11] have produced an immediate release of the chemotherapeutic, which can be risky in the case of using it after resection surgery for pancreatic cancer. The risk is not only due to the possible systemic effects it may have in the recently intervened patient, but also because of the local effects in the area where the device is located such as infection, bleeding, or wound dehiscence. Our study aims to take advantage of the beneficial aspects of the electrospinning technique above-mentioned to overcome the necessity of devices that provide a sustained and delayed release of therapeutic drugs. Along this line of thought, we aimed to fabricate, for the first time, a novel local DDS for pancreatic cancer treatment loaded with irinotecan named *TARTESSUS*. This scaffold provides a delayed release of the chemotherapeutic drug and eliminates any remaining tumor cells in the affected area that were not removed by surgery, preventing in this way the recurrence of the disease [33].

## 2. Materials and Methods

### 2.1. Materials and Reagents for Membrane Fabrication

All of the polymers and solvents used in this project were obtained from Sigma Life Science (St. Louis, MO, USA). Two polymer solutions with different concentrations of PCL in formic acid acetic acid:chloroform (47.5:47.5:5) were used: the first one had 14 w.t.% PCL and the other one 10 w.t.% PCL, being the molecular weight of PCL of 80,000 g/mol. On the other hand, Dulbecco’s phosphate buffered saline (DPBS), purchased from Bioscience Lonza (Morrisville, NC, USA), was used as a medium for drug release. The syringes, heads, and needles were acquired from Becton Dickinson, S.A. (Franklin Jakes, NJ, USA) and the syringe pumps from KD Scientific Inc. Finally, the voltage source used was from Heinzinger (Rosenheim, Germany).

### 2.2. Drug-Loaded Membrane Fabrication (TARTESSUS)

For the fabrication of the membranes, coaxial electrospinning was used, a process where two different solutions can be spun at the same time using an external and an internal flow through the concentric needles that create the jet. Therefore, different polymeric electrospinning solutions were prepared, both loaded or not loaded with the drug. First, formic acid, acetic acid, and chloroform were mixed at 200 rpm for 10 min at room temperature. Then, PCL was added to the mixture in either a concentration of 14 w.t.% or 10 w.t.% and left to mix for 24 h at 1500 rpm and 75 °C. For the solutions containing the drug, after the polymeric solution was homogeneously mixed, irinotecan was added (7 mg/mL) and left overnight at 600 rpm and 75 °C for proper mixing of the solution. As coaxial electrospinning was the method chosen for fabrication, the process provided the opportunity of combining the different solutions in the same process, using one in the inner nozzle and the other in the outer nozzle (Table 1).

After the solutions were prepared, coaxial electrospinning was used to electrospin the nanofibers. The setup was composed of a coaxial nozzle with an inner gauge of 26G and an outer gauge of 18G (Ramé-Hart Instrument Co., Succasunna, NJ, USA), two syringe pumps (KD Scientific Inc., Holliston, MA, USA), and a high voltage DC power supply (Heinzinger, Rosenheim, Germany)To fabricate the fibers, three different configurations were employed. Table 2 shows the parameters for each configuration used. These parameters were initially based on another study, all of them having in common the distance between the needle and the collector of 13.5 cm and the electric field of 21 kV [24].

All three electrospinning configurations from Table 2 (named 1, 2, 3) were performed using the three solution combinations in Table 1 (named A, B, C). Therefore, the nine membranes fabricated with each solution-electrospinning parameter combination were identified as A1, A2, A3, B1, B2, B3, C1, C2, and C3 as a combination of the names of both tables. Membranes with no drug-containing solutions were also fabricated in order to have a “control” for the drug loading and the cell cultures, as explained later.

Once fabricated, meshes were placed in a desiccator for 24 h to remove any residual solvent remaining in the membrane.

### 2.3. Characterization of Nanofibers

#### 2.3.1. Surface Morphology

The surface composition of the samples was evaluated using a scanning electron microscope (SEM, Phenom G2 Pro, Labmate Scientific Inc., Chicago, IL, USA). First, it was necessary to coat the mesh fragments with palladium in an EMITECH^®^ Sputter Coater. Once the images were obtained, the NIH (National Institutes of Health) ImageJ software was used for the fiber diameter analysis. For each scaffold, we measured the diameter of 25 fibers per frame for a total of four images. Thus, in each sample, the diameter of 100 fibers was measured randomly, and the average diameter was reported. Moreover, by image analysis, the nanofiber mats’ porosity as well as the pore size was measured using Diameter J, which is an ImageJ package. The mean value ± standard deviation was used to express the obtained values.

#### 2.3.2. Solvent Remanence

Fourier transform infrared spectroscopy (FTIR) was performed to prove that there was no solvent remanence in the meshes. The equipment used to carry out the analysis was the FTIR Spectrum 100 with ATR (Perkin Elmer, Waltham, MA, USA), with a scanning range between 650 cm^−1^ and 4000 cm^−1^ and a total amount of eight accumulations. FTIR was performed on pure PCL pellets and the solvents used in the dissolutions as well as on the free drug (irinotecan) and the fabricated electrospun meshes. In the case of solid samples (PCL pellets, scaffolds, and free drugs), a force of 90 (arbitrary units of the equipment) was applied. Liquid samples were not compressed.

#### 2.3.3. Drug Loading and In Vitro Drug Release

To evaluate the drug integration inside the fibers, differential scanning calorimetry (DSC, 4000 Perkin Elmer) was used. The weight of the samples used in the characterization was 10 mg. The sweep was carried out from 20 °C to 100 °C at a rate of 10 °C min^−1^, with a one-minute break at 100 °C, and the same conditions for the return to the initial temperature. The initial calorimeter flow used was 20 mWh^−1^. Drug loading was also semi-quantitatively assessed by FTIR analysis of the drug-loaded scaffolds, following the procedure previously described [26], and using the equipment and setups described in the previous section.

Afterward, the determination of the drug release in the different mats was performed by immersing the loaded nanofibers in PBS solution as follows: for each configuration mat, three samples were cut into 3 × 3 cm squares and submerged in 5 mL PBS in a 6-well plate located on an orbital shaker (Polymax 1040 from Heidolph, Germany) [34]. This equipment was introduced in an incubator at 37 °C during the time established for the drug release analysis (30 to 60 days). Spectrophotometry (Eon BiotekTM, Winooski, VT, USA) was used to characterize drug release during this period: 100 µL of the immersing PBS was taken every 24 h for the absorbance reading. With the purpose to analyze the released drug amount, the respective calibration curve was obtained for irinotecan. To calculate drug concentration, Beer–Lambert equations were used, considering λ = 300 nm for the drug. The following equation shows the fitted line for the calibration curve of the irinotecan release in PBS, with a R^2^ value of 0.9858:y = 2.3437x + 0.2639

### 2.4. Determination of Drug Dose per Membrane Surface

To determine the dose per membrane surface, we used the release profiles of irinotecan, and then we calculated the concentration of irinotecan that could be released.

### 2.5. D Cell Culture Conditions

Pancreatic cancer cells PANC-1 (ATCC CRL-1469, Manassas, VA, USA) were cultured in Dulbecco’s modified Eagle’s medium high glucose (DMEM, 12-604F, Lonza, Walkersville, MD, USA) supplemented with 10% (*v*/*v*) fetal bovine serum (F7524, Sigma-Aldrich, St. Louis, MO, USA) and 1% (*v*/*v*) streptomycin/gentamycin (P4333, Sigma Aldrich, St Louis, MO, USA). Then, 152500 PANC-1 cells were cultured in a 6-well plate for 24 h. Cell cultures were treated with 2.25 cm^2^-big *TARTESSUS* (membranes with encapsulated irinotecan). Scaffolds without encapsulated irinotecan were used as controls. We used 300 µL of the medium culture to obtain a concentration of 100 µg/mL of irinotecan, as a previous work reported this value as cytotoxic [35]. Cell viability was measured at 72 h, 7, 10, 14, and 17 days since the beginning of the treatment by performing the PrestoBlue™ HS (P50200, ThermoFisher Scientific, Waltham, MA, USA) cell viability assay.

### 2.6. D Cell Culture

For spheroid formation, 50 µL of 1% (p/v) agarose (A9539-50G, Sigma Aldrich, St. Louis, MO, USA) previously autoclaved was deposited on a 96-well plate to avoid cell adherence on the bottom of the dish. A total of 5000 cells per well were seeded and cultured until day 7, which was the moment when spheroids were stabilized. Spheroids were transferred to a 24-well plate and treated with irinotecan encapsulated in *TARTESSUS*. A membrane without encapsulated irinotecan was used as the control. Cell viability was measured at 72 h and 7, 10, 14, and 17 days after the treatment started. We used 300 µL of the culture medium to obtain a concentration of 100 µg/mL of irinotecan and the medium was not changed during the experiment time to maintain the same irinotecan concentration.

### 2.7. Analyses of In Vitro Proliferation and Apoptosis of Pancreatic Cancer Cells

Apoptosis and cell proliferation were analyzed by immunofluorescence. After removing the medium, cells were washed with PBS (17-516F Lonza, Walkersville, MD, USA) and treated with a 0.1% Triton-X100 solution for 30 s to improve immunofluorescence-staining intensity [36,37]. Then, the cells were fixed with a 4% paraformaldehyde solution for 10–15 min at room temperature, washed with PBS, and stored at 4 °C for 1–2 weeks.

Once this time elapsed, cells were permeabilized with a 0.5% Triton-X100 (A1388, PanReac AppliChem, ITW Reagents, Chicago, IL, USA) for five minutes at room temperature and treated with a blocking solution (1% BSA, 0.1% Tween-20, PBS 1x) for 10 min at room temperature.

Cell cultures and spheroids were incubated overnight at 4 °C in anti-Ki67 (1:200) (SC23900, Santa Cruz, CA, USA) and anti-caspase 3 (1:400) (rabbit mAb 9664S, Cell Signalling, Danvers, MA, USA) primary antibodies. The incubations in goat anti-mouse (488, AB150113, AbCAM, Cambridge, UK) and goat anti-rabbit (555, A21428,ThermoFisher Scientific, Waltham, MA, USA)) secondary antibodies were performed for 1 h, and cells were subsequently stained with DAPI (ProLong P36935, ThermoFisher Scientific, Waltham, MA, USA) (1:1 in PBS) and mounted on glass slides for microscopic examination.

We used a Leica Thunder microscope to take images that were analyzed with ImageJ to quantify the number of cells, the number of positive Ki67 cells, and the number of positive caspase 3 cells. Then, the Ki67-positive cell rate was calculated as the number of positive Ki67 cells over the total number of cells, and the caspase 3-positive cell rate was estimated as the number of positive caspase 3 cells over the total number of cells. Finally, these values were normalized over the 72 h control.

### 2.8. Statistical Analysis

To compare the effect of the solutions and the different electrospinning configurations used in the study on fiber diameter, an independent Student’s *t*-test was performed. When quantitative variables did not show a normal distribution, the Mann–Whitney U test was used for their analysis. For comparison of more than two samples, samples with normal distribution were analyzed using the ANOVA test, while the Kruskal–Wallis test was used for samples with non-normal distribution. The same statistical significance criterion was applied to all the analyses. Depending on the p-value obtained, the statistical difference was determined as follows: not significant, with a *p*-value > 0.05; significant, when the *p*-value was between 0.01 and 0.05; very significant, when the p-value was between 0.001 and 0.01; and finally, extremely significant, when the p-value was lower than 0.001.

## 3. Results

### 3.1. Membrane Characterization

Regarding the membrane characterization, SEM was utilized to observe the nanofibers’ morphology and the diameter for each of the configurations tested in the study. Figure 1 shows the images obtained by SEM, where it can be seen that all configurations produced nanofibers, with a great surface-to-volume ratio. Although no drug crystals were appreciated in any of the obtained images, since all polymeric mats presented homogeneous fibers with different diameters, in some configurations, some deformations such as beads appeared.

According to the fiber size, all fibers were in the nanofiber range as the maximum average diameters were less than 220 µm (Table 3). No significant difference was observed between the fiber diameters obtained with the same solutions and different electrospinning configurations, as all *p*-values were over 0.05. However, a slightly significant difference was observed between the fiber diameter of different solution combinations (*p*-value = 0.045). In fact, on average, the membranes named as A, prepared with a 14 w.t.% of PCL, showed bigger diameters than the ones fabricated with a 10 w.t.% of the polymer (B and C). Porosity was also evaluated, obtaining porosities in the range of 40–65%. The mats with higher fiber diameters were overall less porous scaffolds (Table 3).

On the other hand, after the electrospinning process, the characterization of the compounds that form the fabricated meshes was carried out with infrared spectrum FTIR to ensure that all solvents evaporated during the process. According to the literature, the solvents used in this study presented some peaks. With the FTIR analysis, no residual solvent was found in the membranes after they were desiccated overnight, since the mentioned peaks were not observed in the samples (Figure 2A). However, the peaks corresponding to the presence of the drug were clearly found in the FTIR graph of the drug-loaded scaffolds (around 1500 cm^−1^ and 2000 cm^−1^) (Figure 2B), but not in the pure PCL membranes, demonstrating the drug loading of our fabrication process.

Observing the results of the heat flow curve (DSC) of the membranes fabricated only with PCL for the control analyses, both glass transition temperature (T_g_) and crystallization temperature points could be observed perfectly, whereas the free drug showed a plain line (Figure 2C). As a consequence, membranes of the three types of solution combinations (A, B, and C) containing irinotecan showed temperature peaks much less pronounced than in the PCL-only mat, suggesting that this reduction was due to the presence of the drug in the meshes.

Taking into account that no drug crystals nor deformations on the fibers were observed in the SEM images, and the drug presence observed with both the DSC and FTIR analyses, it was depicted that the drug was successfully encapsulated during the coaxial electrospinning process performed in this study.

### 3.2. Drug Release

When analyzing the release of the meshes of the A-type membranes, in which the concentration of PCL was 14 w.t.%, it was observed that the drug was not released during the first month. Therefore, these membranes were incubated for more time to determine if a late release occurred (Figure 2D). Until day 30, the release curve remained very close to zero and from then on, the drug was released for about 15 days, generating a late liberation. In total, with this type of mesh, more or less 0.5 mg irinotecan was released for 56 days.

On the other hand, the release curve of the B-type meshes (PCL 10 w.t.%) showed a more pronounced release from the eighth day onward. Until approximately the first week passed, the drug was nearly not released and the membrane still maintained more than 90% of the drug. However, from day 8, the absorbance increased, which meant that the drug started to be liberated, generating what was considered a medium release. In this case, the release of irinotecan was 0.45 mg in 30 days.

Finally, C-type membranes, in addition to having the lowest concentration of PCL, also had the drug in both the inner and the outer section of the fibers, so the release was supposed to occur earlier than in the other two cases. After the analysis, an early release of irinotecan was verified: the drug was released from the very first day. In fact, 50% of the drug inside the membrane was released in less than a week

### 3.3. D Cell Cultures

#### 3.3.1. D Cell Viability

Cell viability in 2D cultures showed significant differences between the control and the treated group in all of the measured times of the study (Figure 3A): 72 h (*p* = 0.007), day 7 (*p* = 0.002), day 10 (*p* < 0.001), day 14 (*p* = 0.011), and day 17 (*p* = 0.001).

Within the control group, cell viability was the maximum at 72 h, with a value of 100 ± 4.99%, which decreased at day 7, with a value of 70.94 ± 9.84% (*p* = 0.016) and day 10 (48.74 ± 11.28%) (Figure 3A). Other significant differences were observed as a decrease in cell viability between day 10 (48.74 ± 11.28%) and day 14 (76.60 ± 3.16%) (*p* = 0.020) as well as an increase at day 17 (84.33 ± 11.32%) (*p* = 0.004).

On the other hand, in the group treated with *TARTESSUS*, the cell viability also showed its maximum at 72 h, with a mean value of 70.89 ± 1.16%, and decreased as the experiment elapsed (Figure 3A). Moreover, a significant reduction in cell viability was noticed between 72 h and days 7 and 10 (22.08 ± 6.10%, *p* = 0.040 and −7.04 ± 2.23%, *p* = 0.007, respectively), days 14 (−10.72 ± 4.92%) (*p* < 0.001) and 17 (*p* < 0.001) as well as between days 7 and 10 (*p* = 0.002). However, a significant increase in cell viability was also found in the treated group between days 10 and 17 (*p* = 0.049) and days 14 and 17 (*p* = 0.004).

#### 3.3.2. D Cell Proliferation

Cell proliferation, determined by the Ki67 expression levels (Figure 4), was reduced in the control and treated groups as the experiment elapsed (Figure 3B). At 72 h, cell proliferation of the control group showed a value of 100 ± 44.04%, while in the treated group, this value was 16.89 ± 4.12%. Hence, cell proliferation was significantly lower in the group treated with *TARTESSUS* than in the control group at 72 h (*p* = 0.031).

Additionally, significant differences were observed regarding the Ki67 expression levels between 72 h and day 7 (25.45 ± 24.17%) (*p* = 0.024) as well as between 72 h and day 10 (6.15 ± 1.66%) (*p* = 0.005), day 14 (15.06 ± 2.73%) (*p* = 0.011), and day 17 (26.16 ± 10.84%) (*p* = 0.025).

However, although a significant decrease in cell viability was shown between the control and the treated group on day 14 (*p* = 0.001), with a mean proliferation value in the control group of 15.06 ± 2.73% and 0% in the treated one, no significant differences in cell proliferation were found within the treated group (*p* = 0.527).

#### 3.3.3. D Cell Death

Cell death measured with the caspase 3 expression levels (Figure 4) showed significant differences between the control group and the group treated with *TARTESSUS* at 72 h, with mean values of 100 ± 38.23% and 6659.57 ± 2248.17%, respectively (*p* = 0.037) (Figure 3C). Moreover, significant differences in comparison with the control group at day 10 were also observed, with a value of 20.96 ± 4.56% against the mean value of 1041.29% ± 101.44% observed in the treated group (*p* < 0.001). Similar findings could be seen on day 14 with mean values of 136.47 ± 51.60% in the control group and 915.90 ± 38.58% in the treated group (*p* < 0.001); and on day 17, with mean values of 96.44 ± 55.95% in the control group and 1047.23 ± 222.20% (*p* = 0.002) in the treated one.

Additionally, within the control group, there was a significant increase in caspase 3 levels between days 7 (8.41 ± 2.67%) and 14 (136.47 ± 51.60%) (*p* = 0.014) as well as between days 10 (20.96 ± 4.56%) and 14 (*p* = 0.026). However, no significant differences were observed within the treated group during the established days of the experiment (*p* = 0.073).

Immunofluorescence staining was performed in 2D cultures (Figure 4). The results showed that the cell number gradually decreased in the treated group after the first 72 h after *TARTESSUS* placement. This reduction was also appreciated in pancreatic cancer cell proliferation in the treated group versus the controls, accompanied by an increase in the caspase 3 expression levels. At the end of the experiment, on day 17, very few viable cells were observed in the treated group.

### 3.4. D Cell Culture

#### 3.4.1. D Cell Viability

Statistically significant differences were observed in 3D cultures between the control and the treated group in all the measured times of the study: 72 h (*p* = 0.011), day 7 (*p* = 0.002), day 10 (*p* = 0.002), day 14 (*p* = 0.001), and day 17 (*p* = 0.033). We could also demonstrate that there was a significant difference between viability at 72 h and the rest of the measures: day 7 (*p* < 0.001); day 10 (*p* < 0.001); day 14 (*p* < 0.001); day 17 (*p* = 0.003). However, no significant differences were found within the treated group during the measured times of the study (Figure 5A).

#### 3.4.2. D Cell Proliferation

Cell proliferation of 3D cultures (Figure 6) in the control group only showed significant differences in comparison with the treated group at day 10, with a mean value of 66.68 ± 25.81% and 13.72 ± 10.34% (*p* = 0.030), respectively (Figure 5B).

Additionally, within the control group, significant differences were found in the proliferation levels between 72 h, with a mean value of 100 ± 9.86%, and day 14, with a mean proliferation value of 32.52 ± 16.14% (*p* = 0.020). However, there were no significant differences within the treated group.

#### 3.4.3. D Cell Death

There were no significant differences regarding caspase 3 levels (Figure 5C) between the control and the treated groups, nor differences between the measured times within the control group. However, within the treated group, a significant cell death increase was found between 72 h (76.18 ± 42.28%) and day 17 (402.12 ± 184.82%) (*p* = 0.024) as well as between day 10 (66.92 ± 23.70%) and day 17 (402.12 ± 184.82%) (*p* = 0.020).

#### 3.4.4. Spheroid Size

Spheroid dimensions (Figure 7) in the treated group were significantly different with regard to the control group at day 7 (*p* = 0.002), day 10 (*p* = 0.003), day 14 (*p* = 0.003), and day 17 (*p* = 0.011).

Within the control group, the area of the spheroids increased significantly in comparison with the starting point of treatment, whose area increased up to 127.86 ± 17.50% over their size at the start of treatment. On day 10, the area increased to a value of 529.21 ± 47.98% (*p* < 0.001); on day 14, to a value of 544.79 ± 120.01% (*p* < 0.001); and on day 17, the area increased up to 549.94 ± 100.98% over their size at the beginning of treatment (*p* < 0.001) (Figure 5D).

On the other hand, regarding the treated group, the area showed a significant decrease in comparison to the starting reference state. At 72 h the area increased up to 112.24 ± 20.90, and it decreased to 15.90 ± 5.73% on day 10 (*p* = 0.001), to 4.99 ± 3.07% on day 14 (*p* < 0.001), and to 2.03 ± 1.09% on day 17 (*p* < 0.001).

## 4. Discussion

The major role of any anticancer therapy is to knock down cancer cells to the extent possible with the highest safety for patients. At present, cancer research mainly highlights the management of pancreatic cancer via varied DDSs [38]. In light of the obstacles to the development of effective treatments for pancreatic neoplasms and to reduce the incidence of local recurrence, this work focused on the development of an electrospun nanofiber membrane loaded with irinotecan (*TARTESSUS*) for the post-surgical adjuvant treatment of operated patients.

In the same way as described by other authors [39], depending on the parameters set in the fabrication process, the fibers resulting from the electrospinning process are different, which has a direct impact on drug release. The mats with greater diameter obtained in this study were the ones fabricated with the electrospinning configuration 2, with the three combinations of dissolutions. This, together with the surface volume ratio, led to an improvement in the drug exposition and increased the interaction with the environment by providing a sustained fluid flow across the manufactured scaffold. The results obtained, together with the fiber uniformity noticed in this study, are consistent with the results previously shown by other authors [29].

According to the drug loading characterization of the scaffolds, first, no solvent remanence was found when analyzing the different peaks of the FTIR obtained from the solvents and the electrospun mats. In fact, in addition to analyzing the presence of the solvents, the encapsulation of the drug was also analyzed using FTIR. Although with this technique it is not possible to obtain a numerical result of the encapsulation, previous works have shown that irinotecan presents some peaks between 1500 cm^−1^ and 2000 cm^−1^ [40], which were observed in the drug-loaded scaffolds but not in the ones fabricated purely with PCL. Even more, according to these FTIR results, it was concluded that a higher ratio of inner:outer flow was related to a greater drug concentration on the fibers. When the outer solution did not contain the anticancer drug, a reduction in the outer flow related to the total solution flow induced a greater drug flow, which meant a higher drug encapsulation. This trend was clear in the fibers blended with irinotecan around 1500–2000 cm^−1^ and between 2800 and 3000 cm^−1^, belonging to the C=O stretching and the N–H stretching, respectively. Additionally, when the outer flow was reduced, the processing time was increased, and therefore, the electrospinning process was longer under the same fluid flow of the solution containing the drug. This way, it was observed, for example, in the A-type membranes, that the A1 membrane showed a bigger peak than the A2 one and the A3 one, demonstrating the effect of the flow and the fabrication process time onto the drug loading of the carrier. The fact that the time contact affects the drug loading of the carrier is consistent with other studies [41].

To undertake the structural characterization of the membranes, the DSC equipment was used. According to the literature, PCL has a glass transition temperature (T_g_) of around 20 °C and a crystallization temperature (T_c_) of around 60 °C [42]. Considering that the DSC analysis of the drug on its own showed a plain line, it could be depicted that the A, B, and C membranes had the drug encapsulated on the fibers, and therefore, this was the reason why their heat flow decayed.

Moreover, regarding the pattern release of the A and B type membranes, no similar results were found in the available literature for in vitro studies. However, some authors did find a similar distribution release to the A type on in vivo studies [6,9,13,14,15]. On the other hand, C-type membranes showed an early release of irinotecan. A similar trend was found in another study in which electrospun membranes composed of 2% chitosan fibers showed a gemcitabine release time of 6 days [6] as well as in a similar work with chitosan/sodium alginate composites loaded with doxorubicin [43]. Here, we used irinotecan as part of FOLFIRINOX, the gold standard for pancreatic cancer.

Type B membranes were chosen for our study due to their ability to release irinotecan in a delayed (after 7 days) and sustained way over a close period (30 days). The 7-day delay in the start of drug release may be an advantage in patients operated on for pancreatic cancer since we generated a window of time that will allow for better tissue healing and patient recovery. This overcomes the limitation of type C membranes, in which the drug is completely released in the first week (immediate release) when the priority is to promote patient healing. On the other hand, A-type membranes also provide a delayed and sustained release (drug release was slower), thus reaching lower concentrations with the consequent possibility of reducing the effectiveness of local antineoplastic treatment.

Similar to another study in which gemcitabine-loaded membranes showed a therapeutic effect in pancreatic cancer cells [6], cell viability in our 2D cultures showed a decrease in the treated group with *TARTESSUS* versus the control group. In our case, this decrease was not only significant at 72 h as those authors showed, but also at all the measured times of the study, from 72 h to day 17. A limitation of our study was the maintenance of cell cultures for only 17 days, but since it was not possible to change the cell culture medium and preserve at the same time the initial concentration of irinotecan, the membranes were not kept during the 30 days to prevent cell death due to the lack of essential cell nutrients.

In both the control and *TARTESSUS* treated groups, cell viability showed a decreasing trend from 72 h to day 10, although this decrease was more remarkable within the treated group. In the control group, the initial reduction in cell viability and proliferation could be due to the lack of nutrients, since no medium change was performed for the reasons above-mentioned.

Additionally, the group treated with *TARTESSUS* also experienced a decrease in cell viability during the whole experiment, followed by an increase in the last period. This could be because the cell number at this point was reduced, so any metabolic activity detected hugely affected the cell viability results. Another reason for this behavior could be the development of an adaptative response of the cells in the presence of irinotecan, which allowed some of them to survive and proliferate. According to the available literature, some authors have shown cellular irinotecan resistance in colorectal cancer cells due to an increase in a drug efflux transporter called ABCG2 [44]. However, there was no significant difference between the 72 h measure and the day 17 measure.

Moreover, as the cell viability decreased first, and increased from day 10 onward, so did cell proliferation, measured as Ki67 expression in the control group. However, although at 72 h cell proliferation in the treated group was significantly lower than in the control group, no significant differences were found in cell proliferation within the treated group. Two main facts have to be taken into consideration: (i) that Ki67 is a tumor marker with different roles expressed during the whole cell cycle (except G0) [45], and (ii) that irinotecan targets DNA topoisomerase I, disrupting DNA synthesis and cell replication [46]. Therefore, we can assume that there was a stable expression of Ki67 as cells went through their cell cycle, found that their DNA replication was impeded, and finally died under irinotecan’s influence from day 10 onward. This stable expression of Ki67 as the cells progressed during the cell cycle until they perished due to irinotecan’s mechanism of action explains why there were no significant differences within the treated group. Therefore, we deduced that although there could have been some viability from the cell redox activity detected in the viability assay, there was no cell proliferation.

Following this line of thought, the fact that the spheroids’ size in 3D cultures significantly decreased at the end of the experiment also points in this direction and corroborates our hypothesis that the reduction in cell number was responsible for the increase in cell viability in the final stages of the experiment.

Furthermore, cell viability reduction in the treated group was also accompanied by an increase in the caspase 3 levels compared to the control group, reinforcing the idea that pancreatic cancer cells died in the presence of *TARTESSUS*. However, our results also showed an increase in caspase 3 levels in the control group from day 7 onward, consistent with the cell viability reduction found in this group. This may be due to the baseline levels of apoptosis in our cells during the experiment since no medium change was possible during the procedure.

It is worth mentioning that significant differences in cell viability and proliferation were found between the control and treated groups at 72 h. This is not consistent with the obtained irinotecan release profile, with a starting drug liberation after a week. A possible explanation is that an overdose of irinotecan may cause premature leaking from the membranes, so it might be necessary to regulate the drug dose to prevent its early release. This finding could also result from the presence of the cells and other metabolites, which may cause a premature release of the drug. Regarding this aspect, further research on the release profiles of the electrospun irinotecan membranes in cell culture may be necessary.

To further evaluate the efficacy of *TARTESSUS*, PANC-1 spheroids were used as a tumor model, providing higher complexity to our assessment of the therapeutic effect of our membranes, since spheroids recreate the heterogeneity of pancreatic cancer better than a simplified monolayer culture. The treated 3D cultures showed significantly less viability and proliferation than the controls, and although no differences were found within the treated group in the measured times, the size of the spheroids decreased during the experiment, as another study with a similar approach found out [6]. Moreover, these results are supported by our findings related to the caspase 3 levels, which showed higher spheroid cell death in the treated group at the end of the experiment.

## 5. Conclusions

The electrospun nanofiber membrane (*TARTESSUS*) presented in this study showed the successful encapsulation of a chemotherapeutic drug and a sustained and delayed release with an adjustable release period to optimize the therapeutic effect in pancreatic cancer treatment.

## Figures and Tables

**Figure 1 bioengineering-10-00183-f001:**
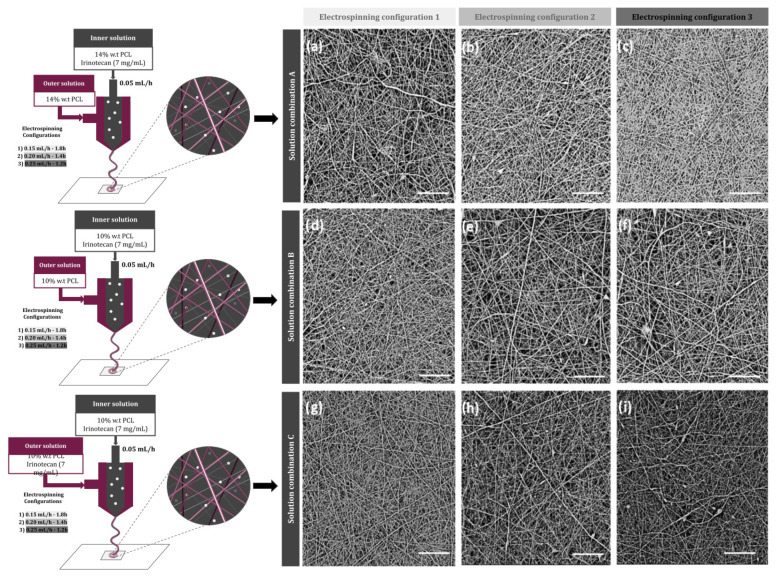
Schematic illustration of the designed electrospun nanofibers and SEM images of different membranes, according to their electrospinning parameters: (**a**) A1, (**b**) A2, (**c**) A3, (**d**) B1, (**e**) B2, (**f**) B3, (**g**) C1, (**h**) C2, and (**i**) C3. Scale bar: 20 µm.

**Figure 2 bioengineering-10-00183-f002:**
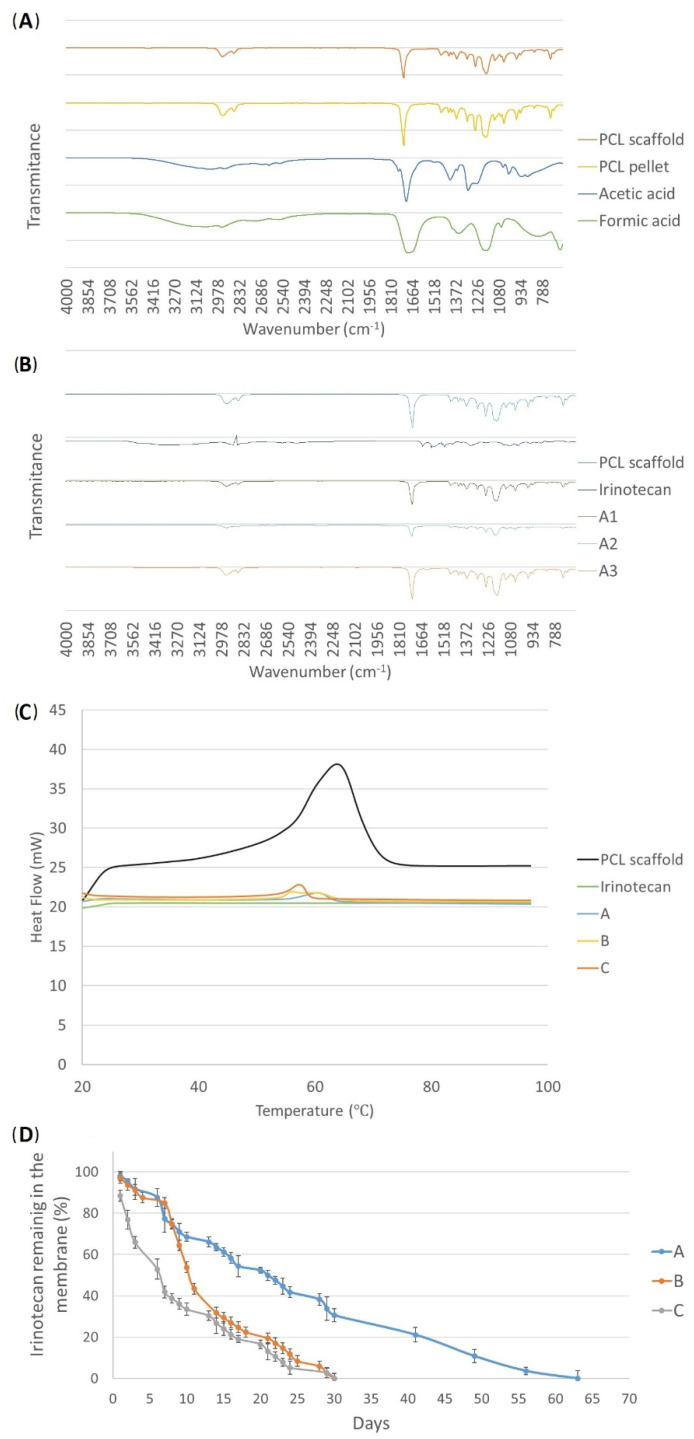
(**A**) Infrared spectrum (FTIR) of the solvents, PCL pellet, and the desiccated PCL scaffold. (**B**) FTIR analysis of the three different “A” type membranes with the different electrospinning parameter configurations, together with the free drug irinotecan and a pure PCL membrane. (**C**) Heat flow curve (DSC) of different membranes (pure PCL and the three solution combinations containing the drug (**A**–**C**) and free irinotecan. (**D**) Release profiles of the electrospun irinotecan membranes according to the manufacturing configurations measured by the spectrophotometer: (**A**) late release; (**B**) medium release; and (**C**) early release.

**Figure 3 bioengineering-10-00183-f003:**
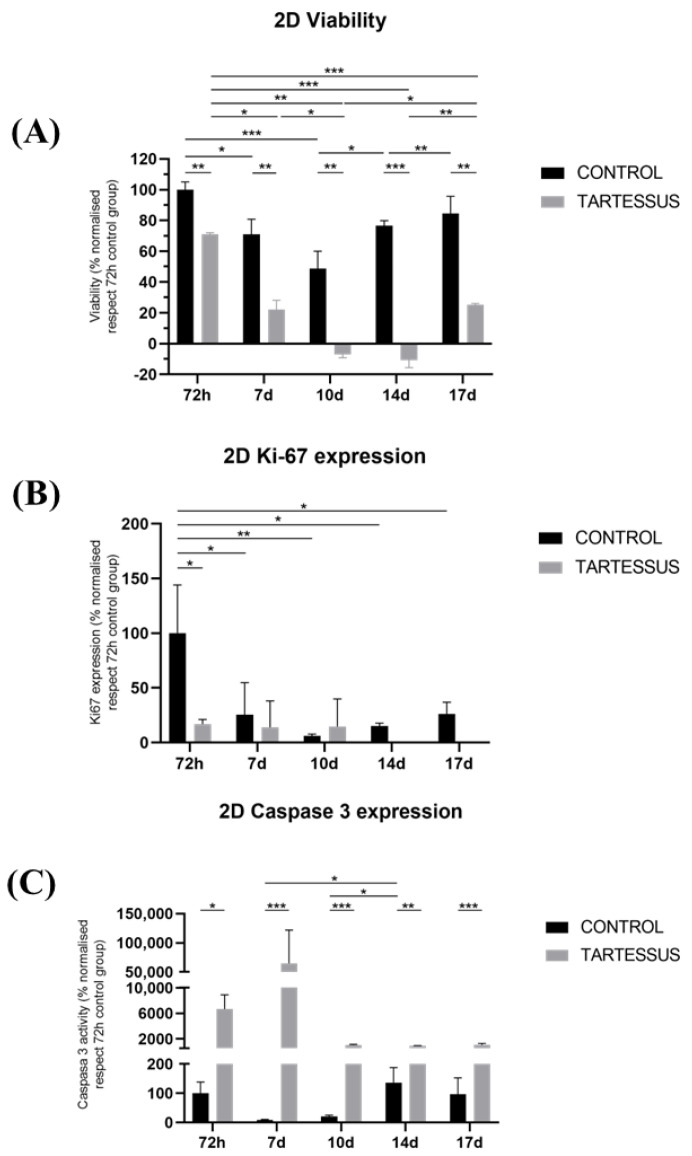
Experimental results obtained from 2D cultures: (**A**) cell viability, (**B**) cell proliferation measured using Ki-67 activity, and (**C**) cell death determined by the caspase 3 levels. Results are shown normalized with respect to the 72 h control group. Significant differences (*), very significant differences (**), and extremely significant differences (***).

**Figure 4 bioengineering-10-00183-f004:**
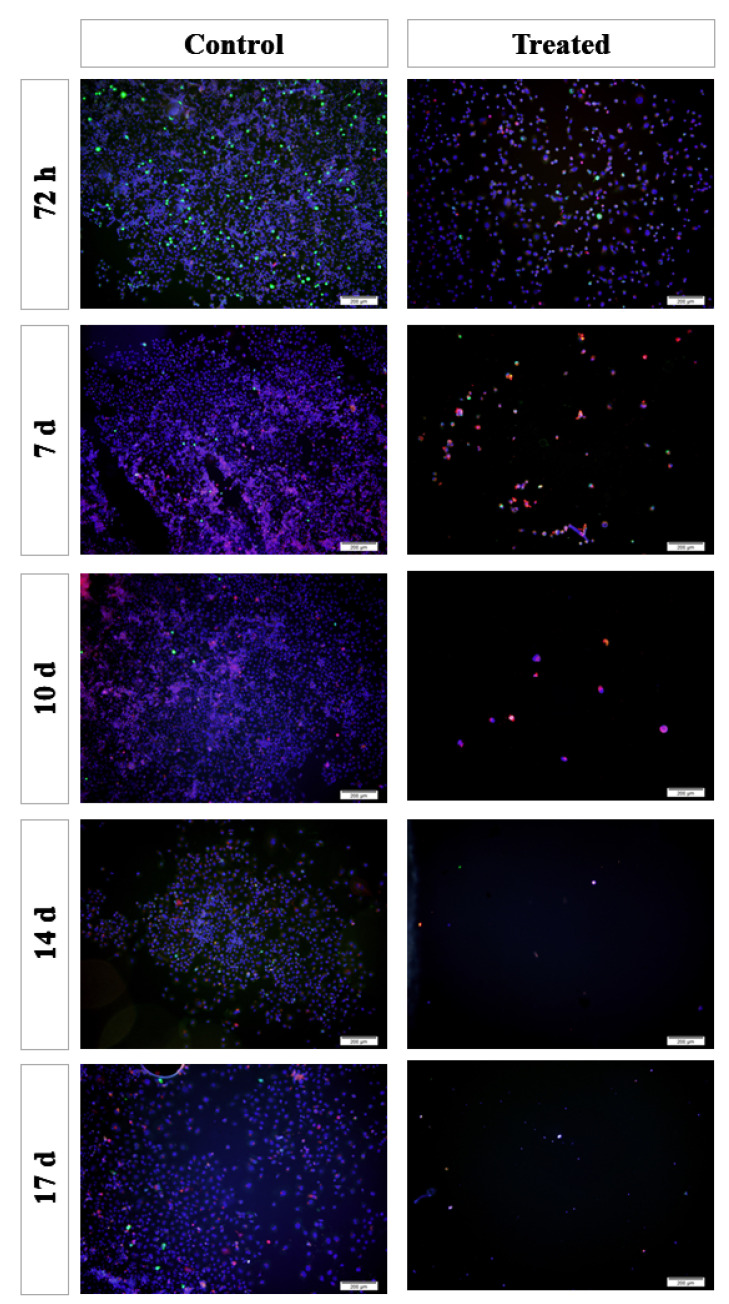
Immunofluorescence staining of the 2D cultures. Blue: cell nucleus (DAPI); red: anti-caspase 3 antibody; green: anti-Ki67 antibody. Scale bar: 200 µm.

**Figure 5 bioengineering-10-00183-f005:**
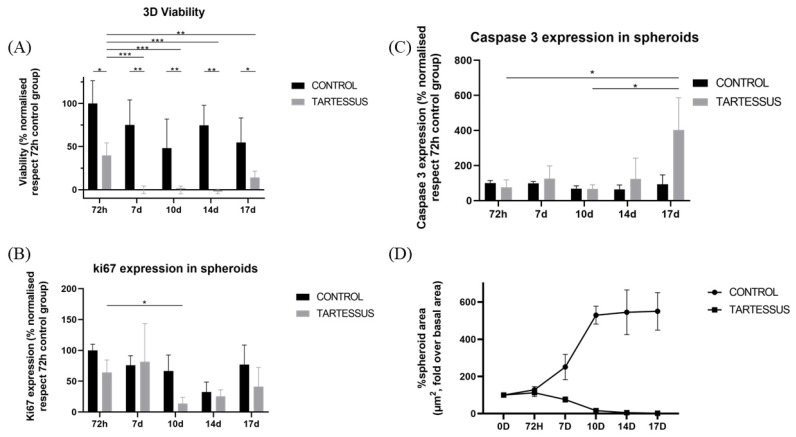
Experimental results obtained from 3D cultures: (**A**) cell viability, (**B**) cell proliferation measured using Ki-67 activity, (**C**) cell death determined by caspase 3 levels, (**D**) spheroid size. Results are shown normalized with respect to the 72 h control group, except for the spheroid size, which is shown normalized with respect to the size at the start of the treatment. Significant differences (*), very significant differences (**), and extremely significant differences (***).

**Figure 6 bioengineering-10-00183-f006:**
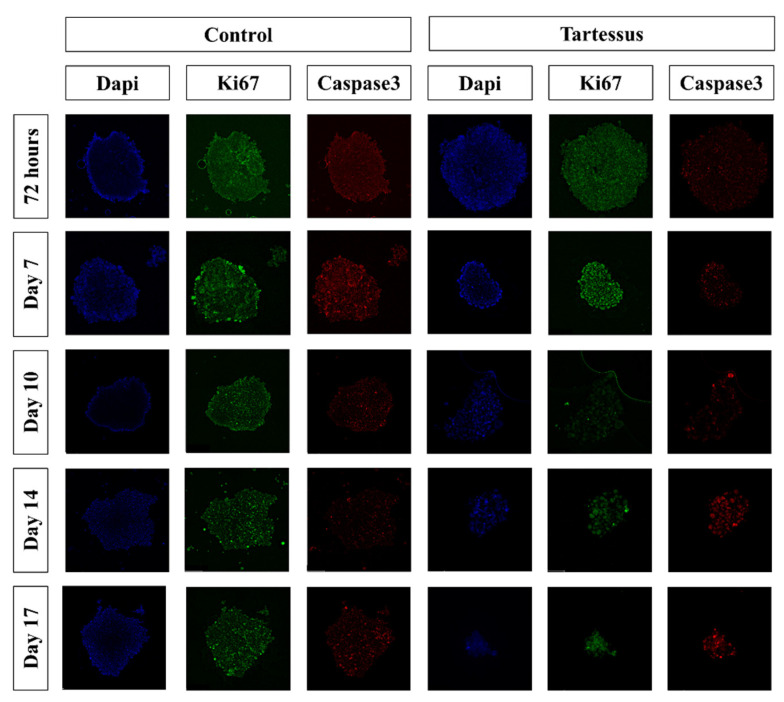
Immunofluorescence staining of the 3D cultures. Blue: cell nucleus (DAPI); red: anti-caspase 3 antibody; green: anti-Ki67 antibody.

**Figure 7 bioengineering-10-00183-f007:**
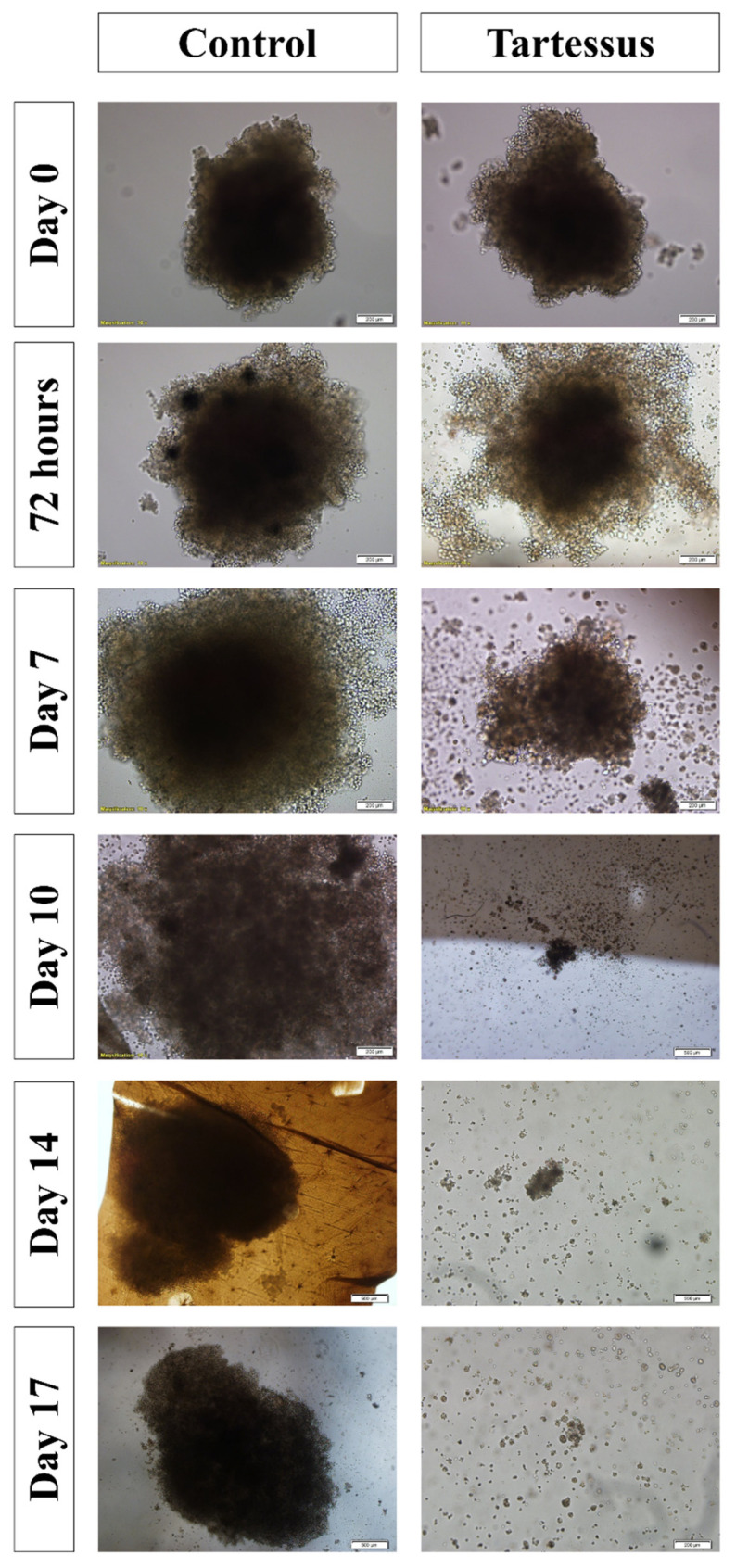
Effect of *TARTESSUS* without and with encapsulated irinotecan. The area of the spheroids (µm^2^, %, fold over control) was measured at 72 h, and on days 7, 10, 14, and 17.

**Table 1 bioengineering-10-00183-t001:** Solutions used in the different coaxial electrospinning processes.

Solution Combination	Inner Solution	Outer Solution
A	14 w.t.% PCL with irinotecan (7 mg/mL)	14 w.t.% PCL
B	10 w.t.% PCL with irinotecan (7 mg/mL)	10 w.t.% PCL
C	10 w.t.% PCL with irinotecan (7 mg/mL)	10 w.t.% PCL with irinotecan (7 mg/mL)

**Table 2 bioengineering-10-00183-t002:** Electrospinning fabrication parameters of the three different configurations for the fabrication of irinotecan-loaded membranes.

Electrospinning Configuration	Collector-Nozzle Distance (cm)	Power Supply (kV)	Outer Flow (mL/h)	Inner Flow (mL/h)	Process Time (hours)
1	13.5	21	0.15	0.05	1.8
2	13.5	21	0.20	0.05	1.4
3	13.5	21	0.25	0.05	1.2

**Table 3 bioengineering-10-00183-t003:** Fiber diameter and porosity of the different irinotecan-loaded membranes using different electrospinning fabrication parameters and polymer solutions in the inner and outer nozzles.

Membrane	Morphology	Diameter (µm)	Porosity (%)
A1	U	0.19 ± 0.04	57.36
A2	U	0.21 ± 0.05	46.25
A3	U	0.18 ± 0.05	64.58
B1	U	0.16 ± 0.03	48.35
B2	U	0.19 ± 0.05	43.97
B3	U	0.18 ± 0.03	60.1
C1	U	0.15 ± 0.03	51.56
C2	U	0.17 ± 0.05	50.65
C3	U	0.14 ± 0.02	53.58

## Data Availability

The data presented in this study are available on request from the corresponding author.

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
