# Peer review of "TARTESSUS*: A Customized Electrospun Drug Delivery System Loaded with Irinotecan for Local and Sustained Chemotherapy Release in Pancreatic Cancer"

_bioengineering, 2023, doi:10.3390/bioengineering10020183_

Round 1

Reviewer 1 Report

In this manuscript, entitled “TARTESSUS: a customized drug delivery system for local chemotherapy in pancreatic cancer”, an electrospun nanofiber membrane loaded with Irinotecan (TARTESSUS) is explored. Obviously, the authors put a lot of effort into this manuscript, some of the technical questions/suggestions are listed below:

1. A revision of the English should be conducted throughout the manuscript.

2. The Introduction needs improvement and rewriting particularly when addressing the aim of the study and the novelty of the work. Please add the latest literature on the novelty and necessity of using electrospun nanofiber membranes as drug delivery systems.

3. In section 2.2, the meanings of “ A1A2A3B1B2…… ” mentioned by the authors are unclear.

4. The quality of all the images needs to be improved, especially figure 2.

5. The data in the manuscript are not sufficient to show that the drug is loaded onto the electrospinning fiber membrane. Please supplement the data.

6. The author mentioned drug load efficiency in section 2.3.3, which was not reflected in the results.

7. Figure 3 is missing from the manuscript.

8. The authors should provide a schematic illustration of the designed electrospun nanofiber membrane.

9. Please check the references again for inconsistencies according to the format of the journal.

10. The very related electrospinning works published recently should to be referred, for example, doi.org/10.1016/j.jece.2022.108908

Author Response

Dear reviewers,

Thank you very much for your comments and suggestions, which we have followed to improve our paper. We respectfully thank you for spending time reviewing our manuscript and for your constructive feedback. Please, appended below you will find a detailed response to your considerations. We have made all the suggested changes and we believe the manuscript has been greatly improved as a result. The changes have been highlighted in the manuscript in yellow. We hope you find them satisfactory to recommend the publication of the work.

In this manuscript, entitled “TARTESSUS: a customized drug delivery system for local chemotherapy in pancreatic cancer”, an electrospun nanofiber membrane loaded with Irinotecan (TARTESSUS) is explored. Obviously, the authors put a lot of effort into this manuscript, some of the technical questions/suggestions are listed below:

  1. A revision of the English should be conducted throughout the manuscript.

Thank you for the suggestion. We have had the manuscript revised by a native English speaker, who guaranteed the English quality that the text has now.

  1. The Introduction needs improvement and rewriting particularly when addressing the aim of the study and the novelty of the work. Please add the latest literature on the novelty and necessity of using electrospun nanofiber membranes as drug delivery systems.

This is a very good point. The Introduction has been modified to ensure that it addresses the aim of the study and the novelty of the work. The aim of the study has been rewritten and the reason behind the necessity of using electrospun membranes has also been addressed in the introduction. In addition, more recent literature has been added also.

  1. In section 2.2, the meanings of “ A1, A2, A3, B1, B2… ” mentioned by the authors are unclear.

We are sorry to read that the nomenclature was not clear from the beginning. Those names are the combination of the solutions in Table 1 with the electrospinning parameters in Table 2. Because of the combination of the parameters in both tables, nine different scaffolds were fabricated. In order to clarify the meaning of the names, the text has been modified as follows: “All three electrospinning configurations from Table 2 (named 1, 2, 3) were performed using the three solution combinations in Table 1 (named A, B, C). Therefore, the nine membranes fabricated with each solution-electrospinning parameter combination were identified as A1, A2, A3, B1, B2, B3, C1, C2 and C3 as a combination of the names of both tables.

  1. The quality of all the images needs to be improved, especially figure 2.

Thank you for the observation. All the images have been improved in order to guarantee a good viewing quality.

  1. The data in the manuscript are not sufficient to show that the drug is loaded onto the electrospinning fiber membrane. Please supplement the data.

Thank you very much for your comment. The drug loading of the scaffolds was demonstrated following the procedures described in the literature, which use DSC and FTIR. The comparison between the empty membranes and the drug-loaded ones in the case of FTIR was missing from the original manuscript, and this was a mistake. Therefore, an explanation has been added in the Results and Discussion sections and Figure 2 has been modified in order to include these data.

  1. The author mentioned drug load efficiency in section 2.3.3, which was not reflected in the results.

Thank you very much for this observation. As mentioned before, the drug loading of the scaffolds was demonstrated following the procedures described in the literature, which use DSC and FTIR. The procedure has been further described in the Methods section (2.3.3. Drug loading and in vitro drug release) and the results have been reflected in the 3.1 section.

  1. Figure 3 is missing from the manuscript.

Sorry for not including the image on the original manuscript. The figure has now been added.

  1. The authors should provide a schematic illustration of the designed electrospun nanofiber membrane.

Thank you very much for the suggestion. In order to improve the understanding of the readers to the electrospun nanofiber membrane design, a schematic illustration has been created and added to the manuscript. In this case, the image has been added as Figure 1.

  1. Please check the references again for inconsistencies according to the format of the journal.

Thank you for the observation. All the references have been checked and the inconsistencies according to the format of the journal have been removed.

  1. The very related electrospinning works published recently should to be referred, for example, doi.org/10.1016/j.jece.2022.108908

This is a very good point. Some recently published works have been referred in the manuscript to improve its quality.

Reviewer 2 Report

A: I hope from the author of this paper to focus about the mathematical models, because of the Mathematical models are an important tool to design pharmaceutical formulations, evaluate drug release processes in vitro and in vivo and, in general, come up with the optimal design for new systems. They allow the measurement of some important physical parameters (e.g., drug diffusion coefficient) and resort to model fitting on experimental release data.

B: What are the types of drug release in this study? Such as; immediate release (IR), delayed release (DR), sustained release (SR), controlled release (CR), stimulus-sensitive release (SSR), and targeted release (TR).

C. The drug can either be incorporated into nanoparticles by hydrogen bonding, ionic interaction, dipole interaction, physical entrapment (or encapsulation), precipitation, covalent bonding or be adsorbed to the surface. In most drug delivery systems, more than one loading mechanism is involved. What are the loading mechanism?

D.  The kinetics of drug release such as; The first order, Higuchi, Korsmeyer-Peppas, and Weibull kinetics models must be applied in this study.

1.  English should be improved throughout the article.

2. I could not found a recognizing contribution in this paper, actually there are a plenty of results about this subject, but the authors were not focusing about the novelty or new information in this field from the papers references.

3. The major results and conclusions must be included in the abstract.

4. The title of the manuscript must be convenient or included the idea of the novelty.

5. Explain in the introduction part the modern method with new carrier in drug delivery system which used in this field such as MCM-41, MCM-48 and SBA-15.

6. Comparative between the characterization of the carrier such as; XRD, SEM, FTIR, BET surface area……etc.

7. The author must be explaining the type of loading in this study. Is the loading was achieved by adsorption method. What about the isotherms of adsorption and kinetics?

Journal of Molecular Structure 1260 (2022) 132879. https://doi.org/10.1016/j.molstruc.2022.132879.

8. I have not found comparative between this study with other literatures.

10. The author must be explaining the role of functional group for prepared carrier.  

Advanced Powder Technology 33 (2022) 103417. https://doi.org/10.1016/j.apt.2021.103417.

11. The loading drug on the carrier is very important. I hope from the author of this paper explain that in details. Drug Delivery, 28 (1) (2021) 856–864. https://doi.org/10.1080/10717544.2021.1914778.

12. The material characterizations for Mesoporous silica material SBA-15 and its derivative SBA@APTES (functionalized with 3-aminopropyltriethoxysilane (APTES)) were prepared and applied as controlled drug delivery vehicles for antibiotic drug cloxacillin (CLOX) must be appeared according the sequence XRD, BET surface area, SEM, FT-IR and TGA.

13. The author must be clear the experimental steps and discuss the interfacial phenomena for this study. 

14. What is the main conclusion from this study? The conclusion must be reduced.

International Journal of Environmental Science and Technology, 2022, 19(3), pp. 1383–1392. https://doi.org/10.1007/s13762-021-03205-5.

15. The surface area is very important. The author must be discussing this point in details because the surface area is very important factor increasing the adsorption rate.

Author Response

Dear reviewers,

Thank you very much for your comments and suggestions, which we have followed to improve our paper. We respectfully thank you for spending time reviewing our manuscript and for your constructive feedback. Please, appended below you will find a detailed response to your considerations. We have made all the suggested changes and we believe the manuscript has been greatly improved as a result. The changes have been highlighted in the manuscript in yellow. We hope you find them satisfactory to recommend the publication of the work.

A: I hope from the author of this paper to focus about the mathematical models, because of the Mathematical models are an important tool to design pharmaceutical formulations, evaluate drug release processes in vitro and in vivo and, in general, come up with the optimal design for new systems. They allow the measurement of some important physical parameters (e.g., drug diffusion coefficient) and resort to model fitting on experimental release data.

Thank you for the comment, this is a very interesting point. We would have loved to apply some mathematical models related to the drug release process in order to obtain the drug diffusion coefficient. However, we are not experts in the field and we have read in the literature that this depends on the drug itself. Therefore, whenever we change the loaded drug inside the fibers, these parameters will change. That is why we focused on the effect of fabrication parameters over drug release, rather than trying to find the most appropriate mathematical model. If the reviewers consider this a must, we will need more time to perform the required experiments to obtain enough data to understand and focus on mathematical model.

B: What are the types of drug release in this study? Such as; immediate release (IR), delayed release (DR), sustained release (SR), controlled release (CR), stimulus-sensitive release (SSR), and targeted release (TR).

This is a good observation. As we mentioned in the Discussion and Conclusion sections of the manuscript, we use different polymeric solutions and electrospinning process configurations to obtain different fibers, with the result of different release profiles. The structure and the content of these fibers is different depending on the combination of the parameters. Type B membranes have the ability to release irinotecan in a delayed (after 7 days) and sustained way over a close period (30 days). The 7-day delay in the start of drug release may be an advantage in patients operated for pancreatic cancer since we generate a window of time that will allow us better tissue healing and patient recovery. This overcomes the limitation of type C membranes, in which the drug is completely released in the first week (immediate release) when the priority is to promote patient healing. On the other hand, A-type membranes also provided a delayed and sustained release (drug release was slower), thus reaching lower concentrations with the consequent possibility of reducing the effectiveness of local antineoplastic treatment.

  1. The drug can either be incorporated into nanoparticles by hydrogen bonding, ionic interaction, dipole interaction, physical entrapment (or encapsulation), precipitation, covalent bonding or be adsorbed to the surface. In most drug delivery systems, more than one loading mechanism is involved. What are the loading mechanism?

Drug loading into nanoparticles is an exciting field. However, we do not work with nanoparticles in this study, but with electrospun fibers. In our case, we dissolve the drug into a polymeric solution. Then, in the electrospinning process, pure polymeric solution and drug-containing polymeric solution are located in the outer and inner part of the nozzle to fabricate a bilayer-electrospun fiber. This way, the drug is physically entrapped into the core (or the outer layer if desired) of the fiber during the electrospinning process.

  1. The kinetics of drug release such as; The first order, Higuchi, Korsmeyer-Peppas, and Weibull kinetics models must be applied in this study.

Thank you for the comment, this is a very interesting point. We would have loved to apply some mathematical models related to the drug release. However, we are not experts in the field and we prioritized other aspects during the elaboration of the manuscript. If the reviewers consider this a must, we will need more time to perform the required experiments to obtain enough data to understand and focus on mathematical models.

  1. English should be improved throughout the article.

Thank you for the suggestion. We have had the manuscript revised by a native English speaker, who guaranteed the English quality that the text has now.

  1. I could not found a recognizing contribution in this paper, actually there are a plenty of results about this subject, but the authors were not focusing about the novelty or new information in this field from the papers references.

We see this point. In order to improve this observation we have tried to show the novelty of the work throughout the manuscript and have compared it to previous studies.

  1. The major results and conclusions must be included in the abstract.

Thank you very much for this suggestion. Both the results and the conclusions have been included in the abstract to provide with a better research summary to the reader.

  1. The title of the manuscript must be convenient or included the idea of the novelty.

Thank you very much for the comment. In order to make the title more convenient and include the idea of the novelty of the work we propose now a new title for our contribution: “TARTESSUS: a customized electrospun drug delivery system loaded with Irinotecan for local and sustained chemotherapy release in pancreatic cancer”.

  1. Explain in the introduction part the modern method with new carrier in drug delivery system which used in this field such as MCM-41, MCM-48 and SBA-15.

Thank you for your comment. We have added in the introduction part some references related to the use of MCM-41, MCM-48 and SBA-15 in the development of drug delivery systems such as nanoparticles.

  1. Comparative between the characterization of the carrier such as; XRD, SEM, FTIR, BET surface area…etc.

Thank you very much for this observation. We would have liked to perform a much broader characterization of the carrier. However, we used all the equipment available on our research center, which were SEM, FTIR, DSC and the spectrophotometer. We know that the use of more techniques (such as the XRD and the BET surface area) would have enriched our outcome, but we were not able to do it, due to a lack of resources.

  1. The author must be explaining the type of loading in this study. Is the loading was achieved by adsorption method. What about the isotherms of adsorption and kinetics? Journal of Molecular Structure 1260 (2022) 132879. https://doi.org/10.1016/j.molstruc.2022.132879

Thank you very much for this comment. The fabrication process explained in our work and the one used in the suggested article are a little bit different. In our case, the carrier is fabricated with a solution that already contains the drug. That is to say, we do not firstly fabricate the carrier and then introduce the drug. Therefore, no adsorption method can be explained in our fabrication method. As explained previously, in our case, we dissolve the drug into a polymeric solution. Then, in the electrospinning process, pure polymeric solution and drug-containing polymeric solution are located in the outer and inner part of the nozzle to fabricate a bilayer-electrospun fiber. This way, the drug is physically entrapped into the core or the outer layer of the fiber during the electrospinning process.

  1. I have not found comparative between this study with other literatures.

Thank you very much for your constructive input. We have tried to increase the amount of references that we cite on the article, comparing their results with the ones we present in our manuscript. We hope that now the quality of the scientific discussion has been improved considerably. 

  1. The author must be explaining the role of functional group for prepared carrier.  Advanced Powder Technology 33 (2022) 103417. https://doi.org/10.1016/j.apt.2021.103417.

This is a very interesting point, which we did not intentionally include in the initial manuscript, but we have discussed in some previous contributions. Therefore, as we also consider that discussing this fact is important, now, the information related to the functional groups that we could observe with the FTIR has been included in the third paragraph of the Discussion section of the manuscript.

  1. The loading drug on the carrier is very important. I hope from the author of this paper explain that in details. Drug Delivery, 28 (1) (2021) 856–864. https://doi.org/10.1080/10717544.2021.1914778.

Thank you very much for this observation. As explained in the suggested article, the time contact has a direct effect on the loading of the carrier. In this experiment, not only the time was important for the drug loading and its subsequent delivery, but also the polymer concentration and their location in the coaxial nozzle played an important role. In fact, as we have now explained more deeply in the Methods section (2.2. Drug-loaded membrane fabrication), in this work, the drug was dissolved in a polymeric solution. First, the polymeric solution was obtained: formic acid, acetic acid, and chloroform were mixed at 200 rpm for 10 minutes at room temperature; then PCL was added to the mixture in either a concentration of 14 w.t.% or 10 w.t.% and left mixing for 24 hours at 1500 rpm and 75 ºC. Once the two different polymeric solutions were prepared, irinotecan was added to them in order to obtain a 7mg/ml concentration of the drug in the final solution. In this case, the drug containing solutions were left overnight at 600 rpm and 75 ºC for a proper mixing. This was vital in order to obtain a homogeneous mixture of the drug with the polymeric solution, as otherwise some clumps of the drug could be observed.

The loading of the drug into the membrane was also different depending on the configuration used during the electrospinning process. As explained in the third paragraph of the discussion, depending on the parameters set in the fabrication process, the fibers resulting from the electrospinning process are different, which has a direct impact on drug loading and release. When the outer solution did not contain the anticancer drug, a reduction of the outer flow related to the total solution flow, induced a greater drug flow, which meant a higher drug encapsulation. Also, when the outer flow was reduced, the processing time was increased and, therefore, the electrospinning process was longer under the same fluid flow of the solution containing the drug. This way, it was observed for example in the FTIR analyses of the A-type membranes, that the A1 membrane showed a bigger peak, than the A2 one and the A3 one, demonstrating the effect of the flow and the fabrication process time onto the drug loading of the carrier.

  1. The material characterizations for Mesoporous silica material SBA-15 and its derivative SBA@APTES (functionalized with 3-aminopropyltriethoxysilane (APTES)) were prepared and applied as controlled drug delivery vehicles for antibiotic drug cloxacillin (CLOX) must be appeared according the sequence XRD, BET surface area, SEM, FT-IR and TGA.

Thank you very much for the observation. However, we do not use the materials mentioned in the comment and we are not sure how the reviewer wants us to compare these materials with our PCL polymer. If further information is provided, we will be happy to improve our manuscript with it.

  1. The author must be clear the experimental steps and discuss the interfacial phenomena for this study.  International Journal of Environmental Science and Technology, 2022, 19(3), pp. 1383–1392. https://doi.org/10.1007/s13762-021-03205-5.

Thank you very much for the suggestion. However, we are sorry to say that we have read the recommended article and have not found the connection between it and our contribution. In the suggested work, nanoparticles zerovalent iron (NFe°) and silty clay supported nano zerovalent iron (SC–NFe°) were utilized as a granular third electrode (3D) in the electrochemical technique. The two electrodes (2D) with aluminum plates as anode and cathode) and 3D electrochemical cells were utilized to remove aqueous phenol. The results demonstrated the considerable efficiency of the 3D electrochemical process in treating phenolic wastewater. Nevertheless, the objective of our study is to fabricate a new drug delivery system. On the other hand, we have tried to clearly explain the experimental steps of this study to improve the quality of the manuscript.

  1. What is the main conclusion from this study? The conclusion must be reduced.

The conclusion has been reduced in order to address the main conclusion of the study. It has been changed to: “The electrospun nanofiber membrane (TARTESSUS) presented in this study showed a successful encapsulation of a chemotherapeutic drug and a sustained and delayed release with an adjustable releasing period to optimize the therapeutic effect in pancreatic cancer treatment.”

  1. The surface area is very important. The author must be discussing this point in details because the surface area is very important factor increasing the adsorption rate.

This is a very good point. We would have liked to perform a much broader characterization of the surface area. However, we used all the equipment available on our research center, which does not have the equipment for a BET surface area analysis. We know that this study would have enriched our outcome, but we were not able to do it, due to a lack of resources.

We were able to measure both the fiber diameter and the porosity of the membranes. What we can say with certainty is that the fact of using the electrospinning process itself increases the surface area of a membrane, as the presence of the nanofibers increases the surface area of the mesh. This, together with the fact that the fibers have irinotecan loaded on their inside, provides a higher drug loading capability than the one that could be obtained just with the coating of a membrane, for example. Therefore, the increased surface area of the scaffolds obtained by the fabrication method presented in this work increases the amount of drug that could be loaded for its delivery.

Reviewer 3 Report

In this manuscript, the authors developed a TARTESSUS with encapsulated Irinotecan, which can delay the release of the chemotherapeutic drug and inhibit pancreatic cancer viability in 2D and 3D models. This is a potentially interesting study. However, there are several major points needed to be addressed:

1, In line 195, “2.4.1. D Cell culture conditions”, 2D or 3D?

2, In line 208 why did the author treat the cell with a 0.1% Triton-X100 solution for 30 seconds before the fix?

3, In line 282, “it was observed that the drug is not released during the first month and, therefore, it was let to release during more time (Figure 2C).”  Please check it.

4, I didn’t see figure3 in the manuscript.

5, In the 2D cell culture, the authors culture the cell for 14 days, I think it is too long, did the authors find the medium turned yellow or change the medium?  The cell viability was reduced on days 7 and 10 proving that the cell was cultured for too long.

6, In figure 5B, why did the Ki67 expression was reduced on day 14? 

7, The authors need to do the animal experiment to prove that the TARTESSUS can eliminate any remaining tumor cells in the affected area 95 that were not removed by surgery.

Author Response

Dear reviewers,

Thank you very much for your comments and suggestions, which we have followed to improve our paper. We respectfully thank you for spending time reviewing our manuscript and for your constructive feedback. Please, appended below you will find a detailed response to your considerations. We have made all the suggested changes and we believe the manuscript has been greatly improved as a result. The changes have been highlighted in the manuscript in yellow. We hope you find them satisfactory to recommend the publication of the work.

In this manuscript, the authors developed a TARTESSUS with encapsulated Irinotecan, which can delay the release of the chemotherapeutic drug and inhibit pancreatic cancer viability in 2D and 3D models. This is a potentially interesting study. However, there are several major points needed to be addressed:

1, In line 195, “2.4.1. D Cell culture conditions”, 2D or 3D?

This is a good point. This doubt has been clarified in the manuscript, by modifying the titles in section 2.5 and 2.6 to show the nature of each of the experiments.

2, In line 208, why did the author treat the cell with a 0.1% Triton-X100 solution for 30 seconds before the fix?

This is a very good point. We used Triton X-100 to improve immunofluorescence-staining intensity as it was reported in previous works such as:

  • Piña, R.; Santos-Díaz, A.I.; Orta-Salazar, E.; Aguilar-Vazquez, A.R.; Mantellero, C.A.; Acosta-Galeana, I.; Estrada-Mondragon, A.; Prior-Gonzalez, M.; Martinez-Cruz, J.I.; Rosas-Arellano, A. Ten Approaches That Improve Immunostaining: A Review of the Latest Advances for the Optimization of Immunofluorescence. Int. J. Mol. Sci. 2022, 23, 1426. https://doi.org/10.3390/ijms23031426
  • Mattei B, Lira RB, Perez KR, Riske KA. Membrane permeabilization induced by Triton X-100: The role of membrane phase state and edge tension. Chem Phys Lipids. 2017 Jan;202:28-37. doi: 10.1016/j.chemphyslip.2016.11.009. Epub 2016 Nov 29. PMID: 27913102.

These two references and the reason why the reagent is used have been included in the manuscript.

3, In line 282, “it was observed that the drug is not released during the first month and, therefore, it was let to release during more time (Figure 2C).”  Please check it.

Thank you very much for this comment. The sentence has been corrected in the manuscript.

4, I didn’t see figure3 in the manuscript.

Sorry for not including the image on the original manuscript. The figure has now been added.

5, In the 2D cell culture, the authors culture the cell for 14 days, I think it is too long, did the authors find the medium turned yellow or change the medium?  The cell viability was reduced on days 7 and 10 proving that the cell was cultured for too long.

This is a very interesting point. The medium culture was not changed during this period in order to keep the same Irinotecan concentration in the treated group. Therefore, with the aim of keeping the same experimental conditions in both groups, the medium was not changed in the control group either. It is true that the medium in the control group turned a little bit yellowish, but nothing remarkable.

6, In figure 5B, why did the Ki67 expression was reduced on day 14? 

This is also a very good point. The Ki-67 expression was reduced in control group on day 7, day 10 and day 14. We think this reduce is produced by the increased of cell number inside the spheroids. Moreover if this cells have no Ki-67 expression, maybe the reduction of the expression over the control at 72 hours is a consequence. Nevertheless, this reduction was not statistically significant.

7, The authors need to do the animal experiment to prove that the TARTESSUS can eliminate any remaining tumor cells in the affected area 95 that were not removed by surgery.

Thank you very much for the suggestion. Indeed, the authors are working on these experiments, which they hope will be published in the future with a larger contribution.

Reviewer 4 Report

The authors composed a manuscript called “TARTESSUS: a customized drug delivery system for local 2 chemotherapy in pancreatic cancer”, which described a local delivery system called TARTESSUS for pancreatic cancer.

1.     Some of the contents in the results section should be moved to the method section (e.g., lines 276-279). The introduction has unnecessary redundancies. The grammar needs to be double-checked. Figure 3 is missing.

2.     Quantification of the ki-67 and Caspase 3 expression level method is missing. Original fluorescence images are missing (Figures 5B and 5C).

3.     Figure 5D requires original images. Why there is a discontinuity in the figure?

4.     The characterization of loading efficiency is missing.

5.     A PBS blank control should be added as a control in the cell experiments.

Author Response

Dear reviewers,

Thank you very much for your comments and suggestions, which we have followed to improve our paper. We respectfully thank you for spending time reviewing our manuscript and for your constructive feedback. Please, appended below you will find a detailed response to your considerations. We have made all the suggested changes and we believe the manuscript has been greatly improved as a result. The changes have been highlighted in the manuscript in yellow. We hope you find them satisfactory to recommend the publication of the work.

The authors composed a manuscript called “TARTESSUS: a customized drug delivery system for local 2 chemotherapy in pancreatic cancer”, which described a local delivery system called TARTESSUS for pancreatic cancer.

  1. Some of the contents in the results section should be moved to the method section (e.g., lines 276-279). The introduction has unnecessary redundancies. The grammar needs to be double-checked. Figure 3 is missing.

Thank you very much for all the observations.

First, the lines 276-279 have been removed from the Results section because it was already explained on the Methods section.

Second, the introduction has been modified in order to improve its quality.

Third, thank you for the suggestion according to the grammar. We have had the manuscript revised by a native English speaker, who guaranteed the English quality that the text has now.

Finally, sorry for not including the image on the original manuscript. The figure has now been added.

  1. Quantification of the ki-67 and Caspase 3 expression level method is missing. Original fluorescence images are missing (Figures 5B and 5C).

Thank you very much for the comment. This mistake has been corrected by adding the next paragraph (together with both images) in section 2.7. Analyses of in vitro proliferation and apoptosis of pancreatic cancer cells:

We used a Leica Thunder microscope to take images that were analyzed with ImageJ to quantify the number of cells, the number of positive Ki67 cells, and the number of positive caspase 3 cells. Then, the Ki67-positive cell rate was calculated as the number of positive Ki67 cells over the total number of cells, and the caspase 3-positive cell rate was estimated as the number of positive caspase 3 cells over the total number of cells. Finally, these values were normalized over the 72 hours control.

  1. Figure 5D requires original images. Why there is a discontinuity in the figure?

This is a very good observation. Some images have been changed or added to Figure 5D so that now there is no discontinuity.

  1. The characterization of loading efficiency is missing.

Thank you very much for your comment. As mentioned before, the drug loading of the scaffolds was demonstrated following the procedures described in the literature, which use DSC and FTIR. The procedure has been further described in the Methods section (2.3.3. Drug loading and in vitro drug release) and the results have been reflected in the 3.1 section. In fact, the comparison between the empty membranes and the drug-loaded ones in the case of FTIR was missing from the original manuscript, and this was a mistake. Therefore, an explanation has been added in the results section and Figure 2A has been modified in order to include these data.

  1. A PBS blank control should be added as a control in the cell experiments.

We see your point. However, we considered that just the PBS could not provide the conditions required as a control. The objective of this study is to analyze the effect of an Irinotecan-loaded PCL scaffold. If we used PBS as a control of the treatment, we would be ignoring the effect that the membrane itself has over the cells. Therefore, by using the plain PCL membrane as a control we guarantee that the effect that we observe over the cells treated with TARTESSUS is actually the effect of the drug that is being released from the fibers over time.

Round 2

Reviewer 2 Report

Thank you very much for achievements all comments in details.

Reviewer 4 Report

The manuscript is accepted for further processing.